# Health care providers'attitude and associated factors to safe abortion in Ethiopia, 2023: A systematic review and meta-analysis

Simachew Animen Bante[1]*, Wondu Feyisa Balcha[1], Fentahun Alemnew Chekole[1], Eden Asmare Kassahun[1], Alemwork Abie Getu[1], Amlaku Mulat Awoke[1], Mengistie Kassahun Tariku[2], Endalamaw Erkie Zerihun[3]

1 Department of Midwifery, College of Medicine and Health Sciences, Bahir Dar University, Bahir Dar, Ethiopia, 2 Department of public health, College of Medicine and Health Sciences, Debre Markose University, Debre Markose, Ethiopia, 3 Department of Obstetrics and Gynecology, College of Medicine and Health Sciences, Bahir Dar University, Bahir Dar, Ethiopia

* animensimachew@gmail.com

## Abstract

### Background

In sub-Saharan Africa, the number of maternal deaths due to unsafe abortions has been gradually rising. In Ethiopia, unplanned pregnancies contribute to 25% of births, accounting for 6%–9% of the maternal deaths resulting from unsafe abortions. Despite several disjointed cross-sectional studies that have been carried out in the past, there is no comprehensive data on the attitudes of healthcare practitioners and other related aspects regarding safe abortion in Ethiopia. This study attempted to measure pooled health care providers' attitudes and determinants of safe abortion in Ethiopia.

### Methods

African Journals Online, Medline/PubMed, EMBASE, Science Direct, Hinari, and Google Scholar were the databases that were accessed. The studies were evaluated critically by using the Joanna Briggs Critical Appraisal methods. The study followed the recommendations set forth by Preferred Reporting Items for Systematic Review and Meta-Analysis (PRISMA). Data were extracted in an Excel spreadsheet and imported to STATA versions 17 software for meta-analysis. The random- effects model was used to pooled the health care providers' attitudes toward safe abortion. Heterogeneity between studies was evaluated using the Cochrane Q-test and $I^2$ statistics (I squared statistics). To evaluate publication bias, egger's tests and funnel plots were employed. Forest plot was used to present the odds ratio (OR) with a 95% confidence interval.

### Results

In this review and meta-analysis, a total of eight papers with a 2,826 sample size were considered. Overall, 65.49% of Ethiopian health care professionals had a positive attitude towards safe abortion (95%CI: 49.64, 81.34; $I^2$ = 99.20%, P = 0.000). Knowledge of the abortion law (OR = 2.25, 95% CI: 1.06, 3.43), being a male provider (OR = 1.89, 95% CI: 1.23, 2.54), receiving training on abortion (OR = 2.91, 95% CI: 1.17, 4.65), working as a

**Data Availability Statement:** All relevant data are within the manuscript and its Supporting Information files.

**Funding:** The author(s) received no specific funding for this work.

**Competing interests:** There is no any competing interest.

**Abbreviations:** CI, Confidence Interval; JBI, Joanna Briggs Institute; KAP, Knowledge Attitude and Practice; MeSH, Medical Subject Headings; MMR, Maternal Mortality Rate; OR, Odds Ratio; PRISMA, Preferred Reporting Items for Systematic Review and Meta-Analysis; VCAT, Values Clarification and Attitude Transformation; WHO, World Health Organization.

midwife (OR = 3.029, 95% CI: 1.605, 4.453) and practicing abortion procedures (OR = 2.55, 95% CI: 1.32, 3.78) were positively associated with the attitudes of the providers regarding safe abortion in Ethiopia.

## Conclusion

In Ethiopia, there was a low pooled prevalence of positive attitude towards safe abortion. Safe abortion services in Ethiopia are more likely to be viewed favorably by health care professionals who have received abortion service training and are familiar with abortion laws. As a result, it is imperative that all healthcare facilities and other relevant parties ensure that health professionals receive training on safe abortion services and are aware of Ethiopia's abortion laws.

## Introduction

Abortion is defined by the WHO as ending a pregnancy before 20 weeks of gestation or if the baby weighs less than 500grams. However, in Ethiopia, abortion is defined as ending a pregnancy before 28 weeks of gestation or if the baby weighs less than 1000grams [1]. Annually, there are approximately 208 million pregnancies worldwide. Among these, unintended pregnancies make up 41%. Worldwide, there are averages of 73.3 million induced abortions (both safe and unsafe) performed each year. Around 80,000 women lose their lives due to unsafe abortions annually, with developing nations bearing over 95% of these deaths and injuries [2]. In numerous countries, particularly in the developing world, limited access to safe abortion services remains a significant contributor to maternal mortality [3]. In sub-Saharan Africa, the incidence of maternal deaths due to unsafe abortions has seen a significant increase. An estimated 30% of women in this area die during pregnancy [4]. Approximately 25% of pregnancies in Ethiopia are unplanned [5]. This contributes 6% to 9% of all maternal deaths each year [6]. Despite the expansion of services following the 2005 reform of the abortion laws, Ethiopia still faces a high rate of complications from unsafe abortions. Nations with high maternal mortality rates, such as Ethiopia, urgently need more interventions and better education for healthcare professionals on safe abortion practices to reduce mortality associated with unsafe procedures [7]. In the federal democratic republic of Ethiopia, the House of Representatives and the Food, Medicine, and Healthcare Administration Control Council of Ministers issued Regulation No. 299/2013, which mandates that "health professionals may not refuse to provide services such as legal safe abortions on the basis of personal beliefs" [8]. Despite this regulation, many medical professionals continue to decline to offer safe abortion services due to their personal moral or religious beliefs [7]. The main obstacles to performing safe abortion procedures in Ethiopia include a lack of facilities and equipment, insufficient knowledge, and the negative attitudes of healthcare workers toward safe abortion [9]. Despite several fragmented cross-sectional studies conducted in the past, there is no cumulative data on the attitudes of Ethiopian health care providers and related factors regarding safe abortion. Therefore, the purpose of this study was to evaluate health care providers' attitudes towards safe abortion and related factors in Ethiopia using a pooled approach.

## Methods

### Study design and setting

The attitudes of Ethiopian healthcare professionals regarding safe abortion were examined through a systematic review and meta-analysis. To conduct and report systematic reviews and

meta-analyses, the study adhered to the Preferred Reporting Items for Systematic Review and Meta-Analysis (PRISMA) guidelines, which consist of checklists (S1 Checklist). Ethiopia, categorized as a low-income country in the Horn of Africa, is projected to have a population of 123.4 million in 2022, 133.5 million in 2032, and 171.8 million in 2050. From an administrative perspective, Ethiopia is divided into 11 regions and two city administrations. These regions are further subdivided into zones, which in turn are divided into districts. Finally, districts are further divided into kebeles, which represent the smallest administrative divisions with a typical population ranging from 2000 to 3500 residents.

## Search strategies and sources of information

We registered this systematic review and meta-analysis in the PROSPERO database with the ID number CRD42022342376. A compressive search from different databases was done including grey literatures. In this review the databases used were Google Scholar, Medline/ PubMed, EMBASE, Science Direct, Hinari and African Journals Online. The search was conducted from February 5 to 10, 2023, and included articles published before February 10, 2023. The search was conducted using four keywords: "Health care provider," "abortion," "factor," and "Ethiopia." For each keyword, relevant Medical Subject Headings (MeSH) terms and entry words were utilized. All the MeSH terms and entry words were combined using "OR" to retrieve a broad range of studies. Then, the four keywords were combined with each of the elements using "AND" to generate specific and relevant articles.

## Inclusion criteria

The main focus of this meta-analysis and systematic review was on articles that met specific criteria. We included studies that reported the prevalence or proportion of health care providers' attitude towards safe abortion and the associated factors. Both published English-language articles and grey literature were considered. We specifically focused on observational study designs, including case-control and cross-sectional studies, that reported the prevalence or proportion of health care providers' attitude towards safe abortion and its associated factors. Published papers or reports up to February 10, 2023, were eligible for inclusion.

## Exclusion criteria

During the selection process, we excluded articles without complete texts or abstracts. Additionally, articles that did not report on the relevant outcome were not included.

## Evaluation of the studies quality

Each study's quality was assessed by using the revised JBI critical appraisal quality assessment tools by using the revised tools for cross- sectional assessment tools [10]. The quality of each original study was independently assessed by all authors using the tools.

## Data extraction

From February 11 to February 30, 2023, all authors utilized a standardized data extraction spreadsheet to gather the necessary data. Each author independently evaluated the tool and reached a consensus. This form includes the author's name, publication year, study setting, study area/region, study design, study period, cases, sample size, providers' attitudes towards safe abortion, and factors influencing these attitudes.

### Outcomes of measurement

The overall pooled percentage of providers with a favorable attitude towards safe abortion in Ethiopia was the primary outcome of the study, whereas associated factors with the attitudes of the providers regarding safe abortion in Ethiopia were the second objective of the review.

### Data analysis

Data were extracted from each study using a format created in Microsoft Excel. For the meta-analysis, the data were imported into STATA 17. The associated factors influencing providers' attitudes towards safe abortion were examined based on eligibility criteria. At least two researches that showed a significant correlation with attitudes about safe abortion were included, taking into account their impact measurements with 95% confidence intervals. Variations between the investigations were evaluated using a random-effects model based on the DerSimonian-Laird technique. Based on the sample size, study region, and study period, sub-group analysis was also carried out. The findings, which included effect sizes and 95% confidence intervals, were displayed using text, tables, and forest plots. Statistical heterogeneity was tested with an $I^2$ statistic at a p-value $\leq 0.05$.

### Heterogeneity and publication bias

Heterogeneity among the included studies was assessed using $I^2$ test statistics and the Cochrane Q-test. The $I^2$ value indicates the percentage of variation across studies that can be attributed to heterogeneity rather than chance. Heterogeneity was considered significant at $p \leq 0.05$. Publication bias was evaluated using funnel plots, and Egger's test ($p < 0.05$) was used to declare publication bias.

## Results

### Description of eligible studies

A total of 1,230 papers were identified through electronic searches. Of these, 400 duplicates were removed, and 800 articles were excluded during the screening process. The remaining 30 articles underwent a full-text review, resulting in 8 articles being considered appropriate and eligible for analysis (Fig 1).

### Characteristics of included investigations

Eight studies met the inclusion criteria and were included in the final systematic review and meta-analysis. The sample sizes ranged from 240 [11] to 431 [12].This systematic review and Meta- analysis consist of eight cross- sectional studies with a total of 2,826 study participants from different regions of Ethiopia. The prevalence ranged from 46.7 [13] to 96.29 [12] (Table 1).

### Providers' attitude to safe abortion in Ethiopia

The overall pooled percentage of providers with a favorable attitude towards safe abortion in Ethiopia was 65.49% (95%CI: 49.64, 81.34; $I^2$ = 99.20%, = 0.000) (Fig 2) with a publication bias distribution of studies was showed by a funnel plot about providers' attitude to safe abortion in Ethiopia (Fig 3).

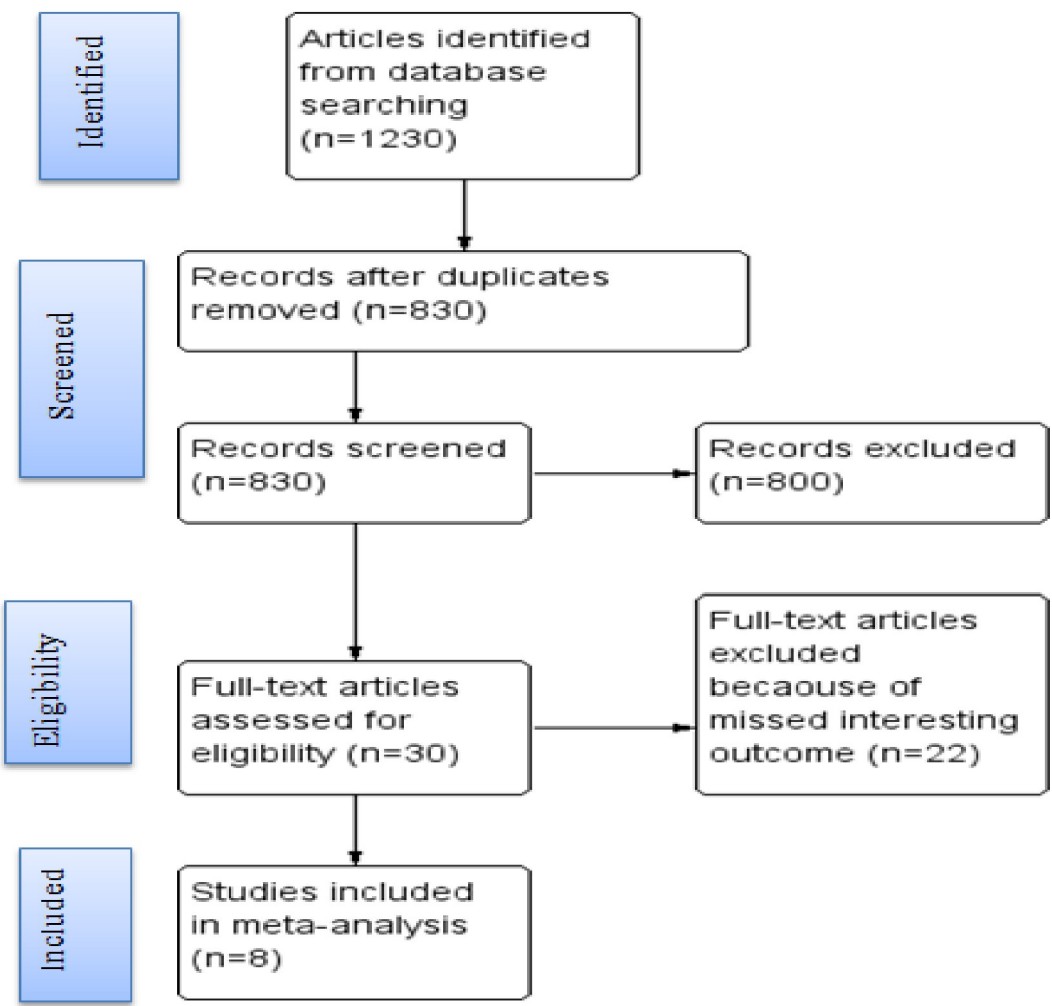

**Fig 1. Flow chart for study of health care providers' attitude and associated factors to safe abortion: A systematic review and meta-analysis in Ethiopia, 2023.**

### Subgroup analysis

Subgroup analysis was conducted based on participant sample size, region, and study period. Although only one study was conducted in the Tigray region, the subgroup analysis indicated that providers' favorable attitudes towards safe abortion were highest in this region, at 89.71%

**Table 1. Study characteristics included in the systematic review and Meta- analysis Ethiopia (n = 8).**

| Authors | Publication year | Study region | Study design | Sample Size | Positive attitude (%) |
|---|---|---|---|---|---|
| Yitagesu S et al [13] | 2018 | Oromia | cross-sectional | 394 | 46.7 |
| Tegenu B et al [30] | 2022 | Harari regional state | cross-sectional | 411 | 58.4 |
| Endalkachew Mekonnen Assefa [31] | 2019 | Adise Ababa | cross-sectional | 405 | 54.1 |
| Terefe T et al [32] | Unpublished | Oromia | cross-sectional | 286 | 53.8 |
| Abebay T et al [33] | Unpublished | Amhara | cross-sectional | 416 | 70.2 |
| Jemila A et al [12] | 2011 | Adise Ababa | cross-sectional | 431 | 96.29 |
| Kalkidan L et al [11] | 2020 | Adise Ababa | cross-sectional | 240 | 54.2 |
| Zaid T et al [34] | 2014 | Tigray | cross-sectional | 243 | 89.7 |

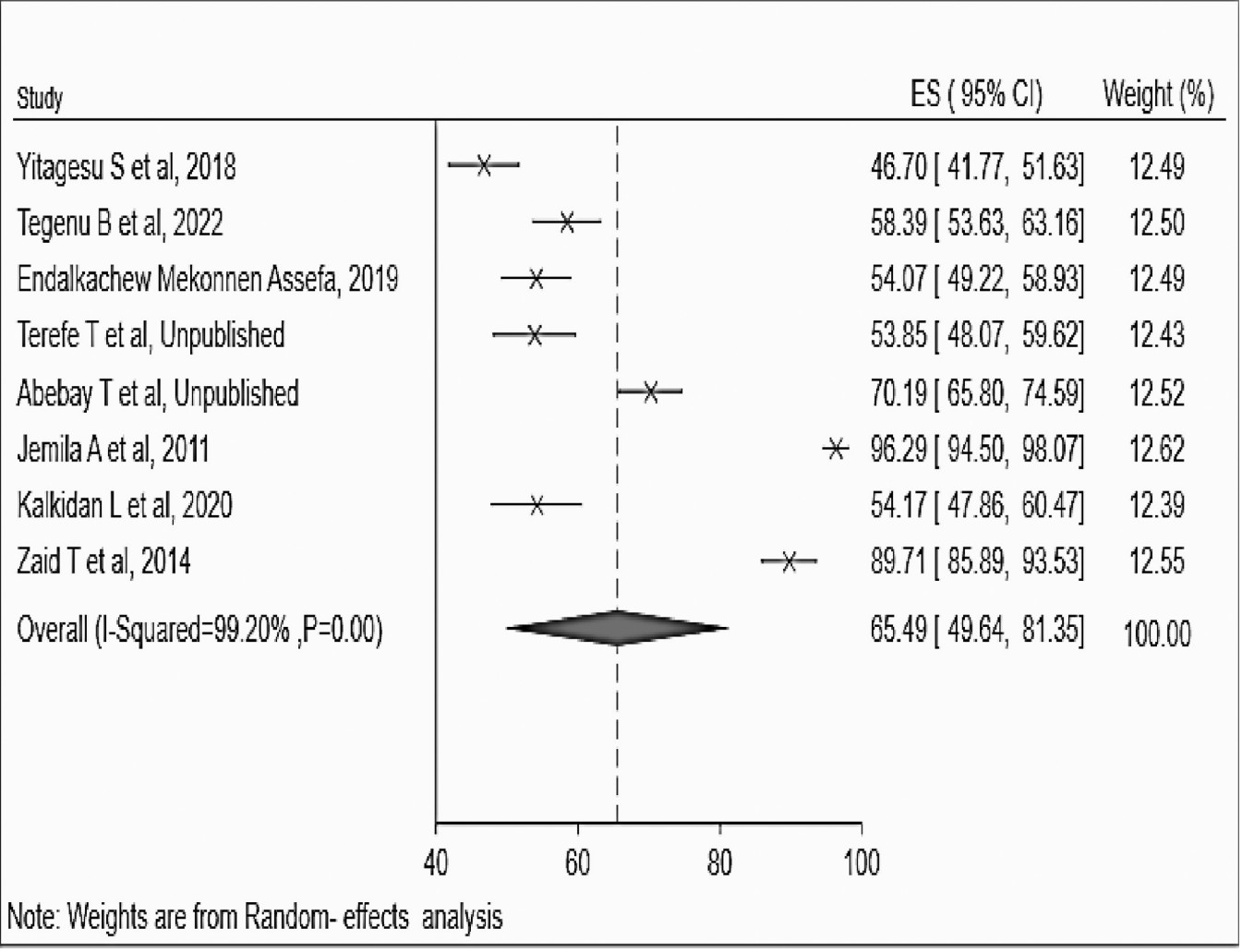

**Fig 2. Forest funnel plot of the pooled providers' favorable attitude to safe abortion in Ethiopia, 2023.**

(85.89–93.53).The pooled prevalence of positive attitudes towards safe abortion is nearly the same across other regions. The subgroup analysis based on sample size revealed that providers' attitudes towards safe abortion were nearly identical. When considering the study period, providers' favorable attitudes were highest in studies conducted in 2016 and earlier (71.76%) compared to those conducted after 2016 (59.32%) (Table 2).

### Factors influencing providers' attitudes towards safe abortion in Ethiopia

This review and meta-analysis identified factors significantly associated with health care providers' favorable attitudes towards safe abortion in Ethiopia. These positively associated factors include knowledge of the country's current abortion laws, being a male health care provider, having received training on safe abortion, working as a midwife, and having experience performing abortion procedures (Table 3).

### Discussion

In this review and meta-analysis, the overall pooled favorable attitude towards safe abortion among Ethiopian healthcare providers was 65.495% (95% CI: 49.640–81.349; $I^2$ = 99.20%,

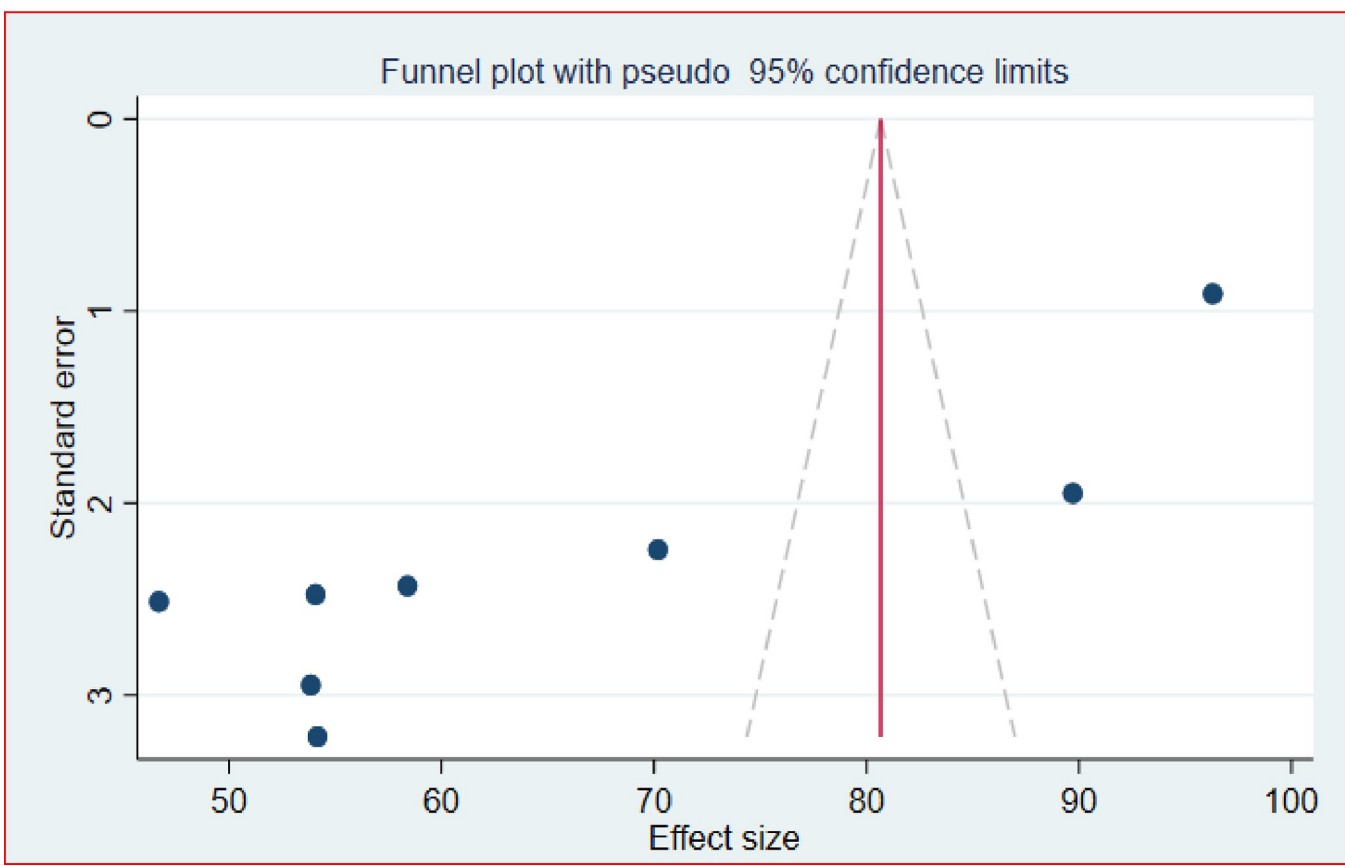

**Fig 3. Funnel plot to test publication bias of eight studies, 2023.**

P = 0.000). The results were consistent with a systematic review conducted in Southeast Asia and Sub-Saharan Africa (70%) [14], as well as national studies conducted in South Africa (70%) [15] and Ghana (80%) [16]. The similarity between these countries could be attributed to their adherence to similar abortion regulations Moreover, healthcare providers in these countries could exchange experiences from international conferences and collaborations

**Table 2. Subgroup analyses to providers' attitude to safe abortion in Ethiopia.**

| Subgroup | Number of study | Sample size | Attitude(95% CI) | Heterogeneity | |
|---|---|---|---|---|---|
| **Region** | | | | I² | P-value |
| Tigray | 1 | 243 | 89.71 (85.89, 93.53) | - | - |
| Amhara | 1 | 416 | 70.19 (65.797, 74.58) | - | - |
| Oromia | 2 | 680 | 50.10 (43.11, 57.10) | 70.61% | 0.000 |
| Harerege | 1 | 411 | 58.39 (53.62, 63.15) | - | - |
| Adise Abeba | 3 | 1076 | 68.25(35.30, 101.21) | 99.47% | 0.000 |
| **Sample size** | | | | | |
| > = 353 | 5 | 2057 | 65.18 (42.81, 87.55) | 99.42% | 0.00 |
| <353 | 3 | 769 | 65.99 (39.88, 92.10) | 98.66% | 0.00 |
| **Period** | | | | | |
| 2016 and before | 4 | | 71.76 (47.89, 95.64) | 99.44% | 0.000 |
| After 2016 | 4 | | 59.32 (51.28, 67.36) | 89.41% | 0.000 |

**Table 3. Analysis of factors associated with providers' attitude to safe abortion in Ethiopia, 2023.**

| Variables | Number of study | Status of heterogeneity | Sample size | OR(95% CI) | $I^2$ | P-value |
|---|---|---|---|---|---|---|
| knowledge of the country's current abortion laws | 3 | Moderate | 1258 | 2.25 (1.06, 3.43) | 44.77% | 0.0002 |
| Being male provider | 4 | Low | 1518 | 1.89 (1.23,2.54) | 28.43% | 0.0000 |
| Training taken on safe abortion | 2 | Low | 827 | 2.91 (1.17,4.65) | 6.96% | 0.0010 |
| Midwives' profession | 2 | No heterogeneity | 821 | 3.02 (1.60, 4.45) | 0.00% | 0.0000 |
| Have ever practiced abortion procedure | 2 | No heterogeneity | 847 | 2.55 (1.32, 3.78) | 0.00% | 0.0000 |

related to safe abortion. They could also receive training on safe abortion practices based on comparable standards. However, this review and meta-analysis reported higher figures compared to studies conducted in Ghana[41.25%] [17],Thailand [41.45%] [18], and in South Africa[38.8%] [19]. Differences in abortion laws and policies across national borders, the impact of abortion on maternal mortality rates, cultural and religious variations, and the nature of the research could all be contributing factors to the observed disparities.

Subgroup analysis by region, sample size, and study period were conducted in our review and meta-analysis. The Tigray region showed the highest positive attitude of health care providers towards safe abortion, with a rate of 89.71% (95%CI: 85.89, 93.53), despite only one study was conducted. The disparity may have resulted from the differences in training accessibility. Regions with better infrastructure and greater training opportunities may have more positive attitudes. Additionally, cultural norms and religious beliefs can significantly impact attitudes toward abortion. Regions characterized by distinct cultural practices or strong religious affiliations may display varying attitudes. Furthermore, the observed difference could be attributed to a single study conducted in the Tigray region.

Based on the findings from this review and meta-analysis, health professionals who were aware of Ethiopia's current abortion laws were 2.25 times (95% CI: 1.066, 3.439) more likely than their peers to have a favourable attitude towards safe abortion in Ethiopia. The results are consistent with research conducted in Southeast Asia and Sub-Saharan Africa [14], Thailand [18], Palestinian [20] and USA [21]. The explanation that could apply is that healthcare providers who are familiar with Ethiopia's current abortion laws can obtain accurate information. This familiarity ensures that they will not face legal consequences for performing the procedure, which may influence their attitudes. Besides, legal abortion laws awareness might contribute to reducing the stigma associated with abortion. When providers know that the law supports safe abortion practices, they may be less judgmental and more empathetic toward patients seeking these services.

Health providers who had received safe abortion training were 2.91 times (95% CI: 1.171, 4.653) more likely to have a favorable attitude towards abortion compared to those who had not received the training. This finding is consistent with studies conducted across twelve countries [22], Thailand [23], a systematic review in the developed world [24], national survey in Ghana [25] and Scotland [26]. This may be because current training packages emphasize knowledge, attitudes, and practices (KAP), and the abortion training provides an opportunity for values clarification and attitude transformation (VCAT) exercise based on real cases. By using real-life cases exercise, the training helps providers shift from a negative to a positive attitude. Moreover, topics on stigmatizing concerns related to safe abortion could be covered in training on the procedure. Thus, this training may help healthcare professionals become more compassionate and nonjudgmental in their approach, which will positively affect attitudes. It may also make them feel obligated to protect patients' rights and well-being, which would positively affect attitudes.

Male providers were 1.89 times (95% CI: 1.237, 2.542) more likely to have a positive attitude towards safe abortion. The findings are consistent with a systematic review conducted in Southeast Asia and Sub-Saharan Africa [14]. This may be due to cultural influences. From an early age, girls are often not permitted to discuss reproductive health issues with their parents or friends in Ethiopia. Additionally, male providers might have greater exposure to discussions about safe abortion practices during their medical training or professional development. This exposure could lead to increased awareness and understanding, positively affecting their attitudes.

When comparing the attitudes of midwives to those of other health professionals, midwives were 3.02 times more likely to be in favor of safe abortion (95% CI: 1.605, 4.453). This finding aligns with research conducted in Colombia [27]. A few potential explanations include training accessibility, working atmosphere, curriculum, and proximity to the subject matter. Midwives frequently provide close prenatal, postpartum, and postpartum care to expectant patients. Because of their experiences, they might come to understand how crucial safe abortion is to maintaining one's reproductive health. They might see it as a means of decreasing risky behaviors and maternal deaths. Furthermore, specialized training in reproductive health, including family planning and contraception, is provided to midwives. Their comprehension of the significance of safe abortion and its function within comprehensive reproductive care may be improved by this instruction.

Compared to healthcare professionals who have never performed a safe abortion, those who have practiced the procedure were 2.55 times more likely to have a positive attitude (95% CI: 1.324, 3.786). This result is supported by research conducted in Thailand [18], Thailand [28] and WHO guideline [29]. Exposure to the procedure may help reduce the anxiety and concern associated with abortion procedure and healthcare professionals might change their perspective from negative to positive attitude to safe abortion. Professionals who have performed safe abortions directly gain practical experience and exposure to the procedure. This awareness might lead to a better understanding of its safety, efficacy, and importance in reproductive healthcare. Additionally, professionals can observe the impact on patients' lives when they are directly involved in abortion care. Being exposed firsthand can increase compassion and empathy, which can result in more supportive attitudes. Besides, the opinions of professionals who have carried out safe abortions may be supported by data and studies. They may realize that lower maternal mortality and better overall results for reproductive health depend on safe abortion practices.

## Limitations

This review is limited to articles published only in English, which may lead to reporting bias.

## Conclusion

In Ethiopia, the pooled prevalence of health professionals who support safe abortion is low. Health care providers' attitudes towards safe abortion in Ethiopia are significantly influenced by their familiarity with the country's current abortion laws, being male providers, having received training in safe abortion techniques, their profession (being midwives), and their experience with performing safe abortion procedures.

To change healthcare providers' attitudes towards safe abortion and increase their awareness of Ethiopia's current laws, it is crucial to offer training that focuses primarily on values clarification and attitude transformation (VCAT) regarding safe abortion. Medical universities and colleges should incorporate Ethiopia's current safe abortion laws into their curricula.

This responsibility lies with the Ethiopian Ministry of Education.

## Supporting information

**S1 Checklist. PRISMA 2020 checklist followed for this systematic review and meta-analysis on health care providers'attitude and associated factors to safe abortion in Ethiopia, 2023: A systematic review and meta-analysis.**
(DOCX)

**S1 Data. Final excel extracted data for this systematic review and meta-analysis on health care providers'attitude and associated factors to safe abortion in Ethiopia, 2023: A systematic review and meta-analysis.**
(XLSX)

## Acknowledgments

The authors would like to thank all the authors and publishers of the primary studies.

## Author Contributions

**Conceptualization:** Simachew Animen Bante.

**Data curation:** Wondu Feyisa Balcha.

**Formal analysis:** Mengistie Kassahun Tariku.

**Investigation:** Alemwork Abie Getu.

**Methodology:** Amlaku Mulat Awoke.

**Resources:** Endalamaw Erkie Zerihun.

**Validation:** Eden Asmare Kassahun.

**Writing – original draft:** Simachew Animen Bante.

**Writing – review & editing:** Fentahun Alemnew Chekole.

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
