## [Decision Letter · Decision Letter 0]

24 Jun 2024

PONE-D-23-39869Health care Providers` attitude and associated factors to safe abortion in Ethiopia, 2023: A systematic review and meta-analysisPLOS ONE

Dear Dr. Bante,

Thank you for submitting your manuscript to PLOS ONE. After careful consideration, we feel that it has merit but does not fully meet PLOS ONE’s publication criteria as it currently stands. Therefore, we invite you to submit a revised version of the manuscript that addresses the points raised during the review process.

ACADEMIC EDITOR: The manuscript titled 'Healthcare Providers' Attitude and Associated Factors to Safe Abortion in Ethiopia, 2023: A Systematic Review and Meta-analysis' has undergone review. Therefore, revisions are required for the introduction, methodology, findings, and conclusion. Extensive language editing is also necessary.=============================

We look forward to receiving your revised manuscript.

Kind regards,

Alqeer Aliyo Ali, MSc

Academic Editor

PLOS ONE

https://assets.researchsquare.com/files/rs-2305231/v1/d5756d6d-f977-469e-a7d1-f8bc3850b2be.pdf?c=1669837351

In your revision ensure you cite all your sources (including your own works), and quote or rephrase any duplicated text outside the methods section. Further consideration is dependent on these concerns being addressed.

Reviewers' comments:

Reviewer's Responses to Questions

Comments to the Author

1. Is the manuscript technically sound, and do the data support the conclusions?

Reviewer #1: Partly

2. Has the statistical analysis been performed appropriately and rigorously? 

Reviewer #1: I Don't Know

3. Have the authors made all data underlying the findings in their manuscript fully available?

Reviewer #1: No

4. Is the manuscript presented in an intelligible fashion and written in standard English?

Reviewer #1: Yes

5. Review Comments to the Author

Reviewer #1: Introduction: The introduction is mentioned very briefly. It is better to explain more about the objectives of the study

Method: 1- Why is demographic and geographic information written in the study design section? Wouldn't it be better if these explanations were explained in the introduction?

2- Prospero code report was enough. Additional explanations in this paragraph should be deleted.

3- In the results measurement section, two variables or two results are written? (It is unclear) only one result is explained.

Discussion: It has not been strongly discussed. Only Ethiopia's cultural problems and education are mentioned. Discussion requires stronger explanations to explain the results.

Conclusion: I think the conclusion is good and practical.

It is better to enter the number of articles of each database in the Prisma flowchart. Not just the total number

The table related to the information of the final 8 articles was not in the manuscript and the supplement file

6. PLOS authors have the option to publish the peer review history of their article (what does this mean?). If published, this will include your full peer review and any attached files.

Do you want your identity to be public for this peer review? For information about this choice, including consent withdrawal, please see our Privacy Policy.

Reviewer #1: No

---

## [Author Response · Author response to Decision Letter 0]

23 Jul 2024

PLOS ONE 

From Simachew Animen

E- mail: animensimachew@gmail.com

Subject: Submitting a Revised Version of the Manuscript

PONE-D-23-39869

Date- 7/19/2024

Title: - Health care Providers` attitude and associated factors to safe abortion in Ethiopia, 2023: A systematic review and meta-analysis 

We would like to thank the reviewers and editiorals for their valuable and constructive comments. We have taken advantage of their indispensable comments to improve our manuscript. In this document, we have described all changes made in accordance with the comments of reviewers. The language issue has been addressed rigorously.We believe that you will satisfy for the revision. Now, the revised manuscript with track changes and unmarked version revised paper without tracked changes are provided in the attached documents.

We are looking forward to hear from you.

With regards!

---

## [Editor Report · Decision Letter 1]

24 Jul 2024

Health care Providers` attitude and associated factors to safe abortion in Ethiopia, 2023: A systematic review and meta-analysis

PONE-D-23-39869R1

Dear Dr. Bante,

We’re pleased to inform you that your manuscript has been judged scientifically suitable for publication and will be formally accepted for publication once it meets all outstanding technical requirements.

Kind regards,

Alqeer Aliyo Ali, MSc

Academic Editor

PLOS ONE
---

## [Editor Report · Acceptance letter]

31 Jul 2024

PONE-D-23-39869R1 

PLOS ONE

Dear Dr. Bante, 

I'm pleased to inform you that your manuscript has been deemed suitable for publication in PLOS ONE. Congratulations! Your manuscript is now being handed over to our production team.

Kind regards, 

on behalf of

Mr. Alqeer Aliyo Ali 

Academic Editor

PLOS ONE